# EscIRL: Evolving Self-Contrastive IRL for Trajectory Prediction in Autonomous Driving

**Siyue Wang**[*1†]**, Zhaorun Chen**[*2]**, Zhuokai Zhao**[2]**, Chaoli Mao**[1]**, Yiyang Zhou**[3]
**Jiayu He**[1†]**, Albert Sibo Hu**[1†]
[1]CIDI Lab, [2]University of Chicago, [3]UNC-Chapel Hill

**Abstract:** While deep neural networks (DNN) and inverse reinforcement learning (IRL) have both been commonly used in autonomous driving to predict trajectories through learning from expert demonstrations, DNN-based methods suffer from data-scarcity, while IRL-based approaches often struggle with generalizability, making both hard to apply to new driving scenarios. To address these issues, we introduce **EscIRL**, a novel decoupled bi-level training framework that iteratively learns robust reward models from only a few mixed-scenario demonstrations. At the inner level, EscIRL introduces a self-contrastive IRL module that learns a spectrum of specialized reward functions by contrasting demonstrations across different scenarios. At the outer level, EscIRL employs an evolving loop that iteratively refines the contrastive sets, ensuring global convergence. Experiments on two multi-scenario datasets, CitySim and INTERACTION, demonstrate the effectiveness of EscIRL, outperforming state-of-the-art DNN and IRL-based methods by 41.3% on average. Notably, we show that EscIRL achieves superior generalizability compared to DNN-based approaches while requiring only a small fraction of the data, effectively addressing data-scarcity constraints. All code and data are available at `https://github.com/SiyueWang-CiDi/EscIRL`.

**Keywords:** Reinforcement Learning, Trajectory Prediction, Autonomous Driving

## 1 Introduction

Trajectory prediction has been one of the crucial components of autonomous system [1], allowing the self-driving vehicle to forecast the movements of itself and nearby entities. Given the rise of modern machine learning (ML) and more specifically deep learning (DL) [2, 3, 4], numerous deep neural networks (DNN)-based methods [5] have been developed for trajectory prediction tasks [6, 7], including more recent works based on Long Short-Term Memory (LSTM) [8, 9, 10] and transformers [11, 12, 13], which take advantage of the more complex and capable DNNs to better capture and understand the intricate temporal and spatial dependencies inherent in various driving scenarios. However, while these approaches can learn a more complex policy representation and are more robust to diverse driving scenarios, their larger-scaled network architectures often require substantial amounts of data to train [14], which poses concerns in underfitting and limits them from being adopted under data-scarce environments [15] or low-resourced computation conditions [16, 17].

Meanwhile, inverse reinforcement learning (IRL)-based approaches [18, 19, 20] have also found great success in predicting trajectories through inferring cost or reward functions from expert demonstrations, and then using these functions to guide the behavior of self-driving vehicles in unseen driving environments. More specifically, given the infinite numbers of potential cost functions that could explain the demonstrations [21], existing IRL approaches typically employ structural priors for these functions, such as a linear combinations of features [22], or constant derivative regularization [23], to reduce the large number of possible cost functions [24].

However, under these assumptions, the learned IRL controllers often suffer from the curse of generalizability and tend to only explain the demonstrations *locally*, making the learned controllers

---

[*]Equal Contribution. † Corresponding Authors: Siyue Wang, Jiayu He, Albert Sibo Hu{wang.sy, he.jy, hu.sibo}@cidi.ai . Work was done during Zhaorun Chen's visit at CIDI Lab.

8th Conference on Robot Learning (CoRL 2024), Munich, Germany.

difficult to adapt to new environments [25]. In other words, the inferred cost or reward functions can only be applied to unseen driving conditions whose types of traffic situations (e.g. left turn at the red light) must have been included in the demonstrations, hurting the system's performance and robustness across a broader range of situations. To handle the real-life complex environments, existing IRL frameworks often pre-classify the mixed-scenario datasets and learn numerous cost functions where each corresponds to one driving scenario. Such rule-based strategies not only increase the unnecessary difficulty in training, but also degrade generalizability, making the system prone to unseen environments and causing safety concerns. As far as we are concerned, no existing sample-based IRL method can learn *global* cost functions for general-purposed trajectory prediction tasks [24].

The limitations of both DNN- and IRL-based approaches highlight the necessity to develop a novel *data-efficient yet robust and generalizable* algorithm for trajectory prediction in autonomous driving. To this end, in this paper we propose **ESCIRL**: an **E**volving **S**elf-**C**ontrastive **IRL** approach that utilizes a decoupled bi-level training framework to efficiently learn generalizable cost function from diverse trajectory distributions without the need of pre-labeled classifications or additional classifiers. More specifically, after initializing with a set of pseudo-controller parameters, on the inner level, ESCIRL employs a self-contrastive IRL (SCIRL) algorithm to increase the likelihood of expert demonstrations within similar distributions, i.e., *positive set*, while reducing expert likelihood in adverse/dissimilar distributions, i.e., *negative set*. And on the outer level, ESCIRL utilizes an evolving framework to iteratively optimize the contrastive sets from the inner level to introduce diversity while ensuring overall convergence. Finally, a continuous controller network (CCN) is trained to capture the mapping from specific trajectory context to the earlier initialized pseudo-parameters.

To summarize, four key contributions of this paper include: 1) introducing ESCIRL, a novel approach that addresses a common challenge in trajectory prediction tasks faced by both DNN- and IRL-based methods, i.e., to ensure generalizability while being data-efficient; 2) a self-contrastive IRL (SCIRL) algorithm that learns a spectrum of robust reward functions from contrasted expert demonstrations without additional labels or classifications, significantly improving generalizability; 3) an evolving framework that iteratively refines the positive and negative sets from SCIRL to enhance convergence while encouraging exploration – we argue that such training framework combining SCIRL and evolving loop may bring benefits to other IRL applications as well; and 4) extensive experiments on two multi-scenario datasets including CitySim [26] and INTERACTION [27], showing that ESCIRL outperforms existing SOTA methods by an average of 41.3%.

## 2   Methodology

In this section, we illustrate the details of our proposed **E**volving **S**elf-**C**ontrastive IRL (ESCIRL). Specifically, ESCIRL aims to efficiently learn a generalizable cost function from diverse trajectory distributions in a self-supervised manner, where it first employs a self-contrastive IRL (SCIRL) algorithm to disentangle similar and dissimilar distributions without requiring any priors. Then ESCIRL adopts an outer-loop evolutionary algorithm to gradually learn the priors via *natural selection* [28] to interpret the multi-scenario distribution for robust trajectory prediction. An overview of the proposed pipeline is shown in Fig. 1. We first summarize the problem statement in §2.1, and then proceed to explain the two major components of ESCIRL, which are the inner-level self-contrastive IRL (SCIRL) module and the outer-level evolving loop in §2.2 and §2.3 respectively.

### 2.1   Problem Statement

Let $\sigma$ and $u$ represent the states and actions of the vehicle respectively, we formulate the dynamics $f(\cdot)$ of each vehicle as:

$$\sigma^{k+1} = f(\sigma^k, u^k). \tag{1}$$

Next, trajectory $\xi$ within the spatial-temporal domain is defined as a sequence of states and actions, i.e., $\xi = \{\sigma^0, u^0, \sigma^1, u^1, \ldots, \sigma^{N-1}, u^{N-1}\}$, where the superscripts indicate the discretized planning step and $N$ denotes the total length of the planning horizon.

Given a set of driving demonstrations $\mathcal{D}_p = \{\xi_i\}$ comprising of various individual trajectories as $i = 1, 2, \ldots, N$, and a set of parameters $\omega$ of a reward function $R$, the probability of the expert

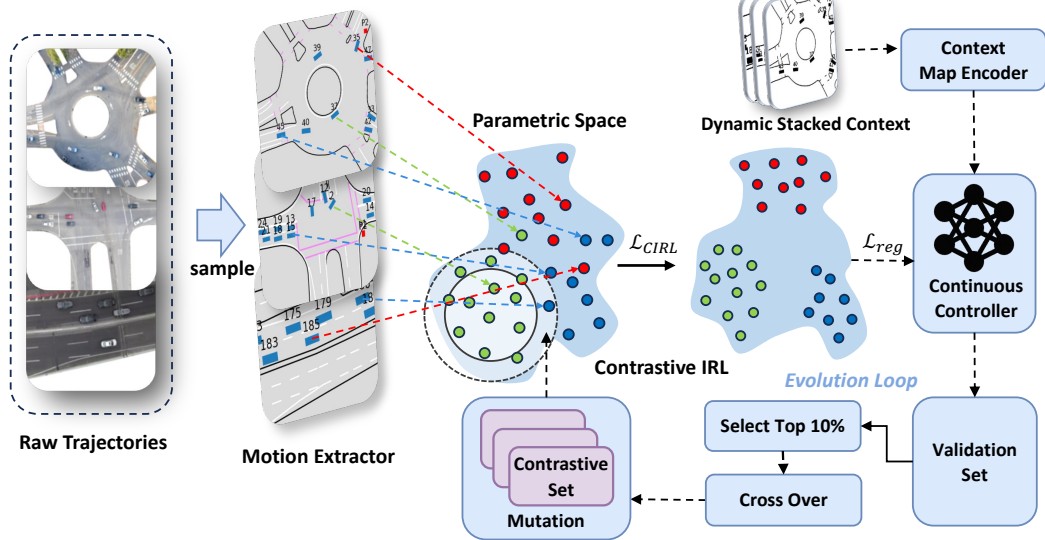

Figure 1: EsCIRL overview. Given a raw trajectory dataset, we first converts it into Frenet coordinates using a low-dimensional motion extractor. Next, we initialize its population by randomly sampling a pair of positive and negative set for each trajectory entry in the dataset. Then, we apply scIRL to each contrastive set, and obtain a set of IRL parameters w.r.t. each trajectory in the dataset. With the set of IRL parameters, EsCIRL fits a unified continuous control network (CCN) to adaptively predict control parameters given the dynamic stacked context (e.g. vehicle states, map geometry) and select the top 10% of the population for crossover and mutation to produce the subsequent population. This evolving process is repeated for a maximum of $K$ times or until convergence.

demonstrations $\mathcal{D}_p$ is commonly [29] defined as:

$$P(\mathcal{D}_p \mid \omega) = \frac{1}{Z_\omega} e^{\beta R_\omega(\mathcal{D}_p)} = \frac{1}{Z_\omega} e^{\beta \omega^T \mathcal{F}(\mathcal{D}_p)} \approx \prod_{i=1}^{N} \frac{1}{Z_\omega} e^{\beta R_\omega(\xi_i)} \qquad (2)$$

where $Z_\omega$ denotes the set of feasible trajectories, and $\beta$ specifies the demonstration proximity [30], which is set to 1 throughout this paper unless otherwise noted. The goal of an (maximum entropy) IRL algorithm [29] is to infer the underlying reward function that maximizes Eq. (2), which is the likelihood of all expert demonstrations, with the assumption that trajectories are exponentially more likely with higher cumulative rewards [31]. More specifically, assume that the trajectory space can be approximated with discretized samples $Z_\omega \approx \sum_{m=0}^{M} e^{\beta R_\omega(\xi_i^m)}$, where $\xi_i^m \in \mathcal{D}_m^i$ is a candidate trajectory in the sampled trajectory set $\mathcal{D}_m^i$ for demonstration $\xi_i$. Thus the goal of IRL is to find a set of parameters $\omega^*$ which maximizes the expert demonstration among the set of sampled trajectories. Mathematically, we have

$$\omega^* = \arg\max_\omega \frac{1}{N} \log P(\mathcal{D}_p|\omega) = \arg\max_\omega \frac{1}{N} \sum_{i=1}^{N} \log P(\xi_i|\omega) \qquad (3)$$

In the context of trajectory planning and prediction, $\omega^*$ will then be used to select the best trajectory from the sampled trajectory set $\xi_i^* \in \mathcal{D}_m$.

## 2.2 Self-Contrastive IRL

Following [19, 32], we employ a linear-structured reward function with respect to $\mathcal{F}$, which is a chosen feature space. Specifically, we follow [32] and define three categories of features separately for *efficiency*, *safety* and *comfort*. Besides, we propose two novel interactive features to capture the driver's desiderata in complex intersections. Finally, we define twenty-two features in total, more details on the feature design are in Appendix C.

While existing works typically partition datasets first into specific scenarios that are clearly distinctive from each other such as dividing the dataset into subsets of in-lane following [22], lane-

changing [32], or lane-merging [33], so that the inferred reward function ($w^*$ in Eq. (3)) is stable across the demonstrations within the *in-domain* scenario, we instead consider the complete *multi-distribution* dataset without any partition to learn a unified controller network that outputs such $w^*$ to handle all the diverse driving scenarios. Learning such unified controller network brings more challenges than simply solving for Eq. (3), which produces only a single vector of the control parameters. Notice that this is fine for existing works as they have an implicit assumption that the expert demonstration obey a unified distribution [22] due to partition. However, such assumption is no longer valid in our case as we consider multi-distributions in the expert demonstrations. Blindly following such assumption would lead to decreased performance and lower robustness [24]. Therefore, to handle the multi-modality in terms of driving conditions and preserve the distribution diversity in the demonstration dataset, we propose a self-contrastive IRL (sCIRL) algorithm. Inspired by [34], we augment Eq. (2) with an auxiliary contrastive loss term, which can be represented as:

$$\mathcal{L}_{\text{sCIRL}}(\omega) = \mathcal{L}_{\text{IRL}}(\omega) + \gamma \mathcal{L}_{\text{Contrastive}}(\omega) \tag{4}$$

where $\gamma$ is a hyper-parameter and is set to 1 throughout this paper unless otherwise noted. Different from [34], sCIRL considers both positive (similar policy) and negative (dissimilar policy) set, with $\mathcal{L}_{\text{IRL}}$ only being conducted on the positive set, i.e.

$$\mathcal{L}_{\text{IRL}}(\omega) = -\frac{1}{K^+} \sum_{i=1}^{K^+} \left( \omega_i^T \mathcal{F}(\xi_i) - \log \sum_{m=0}^{M} e^{\omega_i^T \mathcal{F}(\xi_i^m)} \right) \tag{5}$$

and we design $\mathcal{L}_{\text{Contrastive}}$ as a hinge loss that seeks to ensure that the expert demonstration has a higher probability in the positive set than the negative set by a margin $\epsilon$. Mathematically, we have:

$$\mathcal{L}_{\text{Contrastive}}(\omega) = \max \left( P^- - P^+ + \epsilon, 0 \right) \tag{6}$$

where $P^+$ and $P^-$ are the probability of each expert demonstration in the positive and negative set respectively. More specifically, for an arbitrary demonstration $\xi \in \mathcal{D}_p$, we have

$$P_j^+ = \frac{1}{K_j^+} \sum_{i=1}^{K_j^+} \frac{e^{\omega_i^T \mathcal{F}(\xi_i^+)}}{\sum_{m^+=0}^{M} e^{\omega_i^T \mathcal{F}(\xi_i^{m^+})}}, \quad P_j^- = \frac{1}{K_j^-} \sum_{i=1}^{K_j^-} \frac{e^{\omega_i^T \mathcal{F}(\xi_i^-)}}{\sum_{m^-=0}^{M} e^{\omega_i^T \mathcal{F}(\xi_i^{m^-})}} \tag{7}$$

where $m^+, m^-, \xi_i^+$, and $\xi_i^-$ represent the demonstration $m$ and sampled trajectory $\xi$ in the positive $(+)$ and negative $(-)$ subset for trajectory $\xi_j$. $K_j^+$ and $K_j^-$ denote the number of demonstrations in the positive and negative set. Based on a common assumption that each demonstration $\xi_i \in \mathcal{D}_p$ only represents one certain scenario, we define a separate pair of contrastive set for each $\xi_i$. In other words, instead of applying $\mathcal{L}_{\text{IRL}}$ on the overall dataset as in Eq. (2), we define a specific pair of positive/negative (contrastive) sets for each demonstration $\xi_j$ in the dataset $\mathcal{D}_P$ More details are illustrated in Algorithm 1. As a result, sCIRL can effectively preserve the multi-modality (multi-driving conditions) across diverse driving scenarios and optimize expert behaviors on a more fine-grained subspace. However, considering the general case where no priors are available to pre-determine these sets, we introduce an evolutionary algorithm to iteratively refine the contrastive sets in a self-supervised manner.

## 2.3 Evolving Framework

Since sCIRL does not rely on any priors to pre-define the contrastive sets for each trajectory $\xi_j$, we propose an evolving framework, which is considered as the outer-level of EsCIRL integrating sCIRL into an iterative closed-loop optimization. In this way, we can effectively accumulate the priors to determine contrastive sets for each trajectory via *natural selection* [28], so that convergence and global optimality can be guaranteed. As shown in Fig. 1, for each raw trajectory, we first transform it to the Frenet state-space [35] using a rule-based motion extractor. Next, for each instance in the initial population, we set $\xi_i^+ = \xi_i$ and $\xi_i^- = \varnothing$ for each trajectory in the dataset and then apply sCIRL to obtain a set of corresponding reward parameters $\omega_i$ as initialization.

**Continuous control network.** To capture the underlying relationship between a specific driving context including vehicle states (both spatial and kinematic), map geometry, and its expert policy

---

**Algorithm 1** Updating CCN with EsCIRL

---

**Require:** a dataset $\mathcal{D} = \{\xi_i\}_{i=1}^N$ with $N$ trajectories $\xi_i$; feature function $\mathcal{F}$; safety-constrained sample policy $\pi_i$; number of points in contrastive set $K^+$, $K^-$, learning rate $\alpha$, convergence threshold $\epsilon$

**Ensure:** a global cost function $G$; sample policy $\pi^*$

1: Randomly initialize $\omega_i, \pi_i$ for all $i \in \{1, \ldots, N\}$
2: Project trajectory $\xi_i$ into feature space: $F_i = \mathcal{F}(\xi_i)$
3: **for** each trajectory $\tau_i, i \in \{1, \ldots, N\}$ **do**
4:      Sample $M$ trajectories under policy $\pi_i$
5:      Find $K^+$ closest and $K^-$ furthest points by sorting distance matrix $\{d(\omega_i, \omega_j)\}_{j=1}^M$
6:      Compute contrastive IRL loss $\mathcal{L}_{\text{sCIRL}}(\omega_i)$ for $\omega_i$:
7:      Update $\omega_i$ using gradient descent: $\omega_i \leftarrow \omega_i - \alpha \nabla_{\omega_i} \mathcal{L}_{\text{sCIRL}}(\omega_i)$
8:      Update $\pi_i$ using policy gradient (if $\pi_i$ is learnable)
9: **end for**
10: Fit CCN to predict $\{\omega_i\}_{i=1}^N$ from the initial state $\{\sigma_i^0\}_{i=1}^N$ of trajectory $\{\xi_i\}_{i=1}^N$
11: Predict reward parameter $\omega_i^{pred}$ with CCN
12: Update CCN using gradient descent with loss function $\mathcal{L}_{\text{CCN}}(\omega_i^{pred}|\omega_i)$

---

distribution, we propose to learn a unified continuous control network (CCN) to adaptively predict the corresponding parameters of the reward function (as in Eq. (2)) for the ego vehicle. The architecture of the proposed CCN is shown in Fig. 2a, with three streams of network inputs: the static map geometry, mapping of vehicle spatial states (e.g. coordinate, heading), and mapping of vehicle kinematic states (e.g. velocity, accelerations, jerk). The output is the corresponding control parameters $\omega_i$ of the reward function.

Since both the map geometry and vehicle spatial states are static, we stack them to be a dual-channel mapping input. Then we design a Siamese structure [36] in CCN to separately encode the vehicle kinematic information (i.e. kinematic input), and the static map with vehicle spatial information (i.e. spatial input). Notably, we define the intermediate mapping output of the convolution for spatial input to be an attention map [37], indicating which specific parts of the context are more responsible for the current predictions. On the other hand, the convolution for kinematic input encompasses a richer representation of the vehicles' dynamic states. After obtaining both convolution outputs, we fuse the two mappings using a Hadamard Product and then concatenate it again with the ego vehicle's states to further augment the low-dimensional representation of vehicle-environment information. Finally, the concatenated vector is mapped to the sCIRL parameters space through a fully-connected layer. During training, the loss of CCN is defined as:

$$\mathcal{L}_{\text{CCN}}(\omega^{pred}|\omega) = \frac{1}{N} \sum_{i=1}^N \log(\cosh(|\omega_i^{pred} - \omega_i|)) \tag{8}$$

where $\omega_i^{pred}$ and $\omega_i$ refers to the predicted reward parameter and ground truth respectively. And $\cosh(x) = \frac{e^x + e^{-x}}{2}$. In general, we argue that this attention-based design allows for an adaptive representation of the driving context and establishes an informative mapping from the driving context to the controller space (i.e. parameter space of reward functions). Furthermore, we provide an analysis on interpretability for the proposed CCN structure in §4.4. The procedure for updating CCN within EsCIRL is detailed in Algorithm 1.

**Evolving contrastive sets.** As shown in Fig. 1, after sCIRL converges in each iteration, we select the top 10% of the population based on averaged displacement error (ADE) from the in-domain validation set, and further crossover and mutate them to produce the subsequent populations. Specifically for crossover, we sample $I_E$ mutual demonstrations from the intersection set of the top 10% contrastive sets and sample $(K^{+/-} - I_E)$ demonstrations from the difference set of the union and intersection sets. Then we mutate the demonstrations in each contrastive sets by a uniform probability of 5% to obtain each population in the next generations. This process is repeated until convergence (the decrease of ADE falls below a threshold) or for a maximum $K$ times, as shown in Fig. 2b.

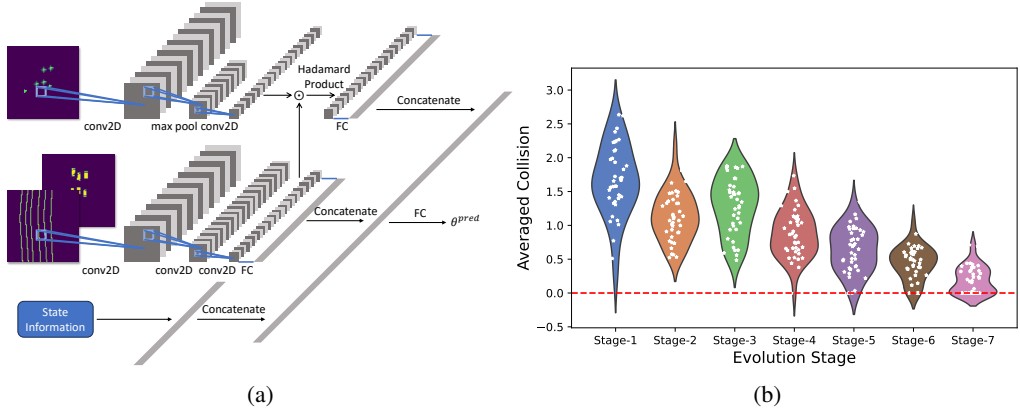

|     (a)     |     (b)     |

Figure 2: **(a)**: we first encode the vehicle state $\sigma_i$ into a multi-channel context map $\mathcal{M} \in \mathcal{R}^{c \times \mathcal{W} \times \mathcal{W}}$ with $c$ channels of size $(\mathcal{W}, \mathcal{W})$ to separately involve the spatial and kinematic information of the vehicles and surrounding map geometry. **(b)**: the white dot represents a candidate of the contrastive pairs, where each corresponds to a policy; the y-axis shows its corresponding averaged number of collision after convergence. The red dashed line indicates zero averaged collision.

## 3    Experiment

**Datasets.** To evaluate the capability of ESCIRL on complex multi-scenario driving modeling tasks, we use INTERACTION [27] and CitySim [26] datasets. INTERACTION is a comprehensive dataset of interactive vehicle trajectories collected from various traffic scenarios (e.g. roundabout, intersections) around the world. And CitySim is a drone-based vehicle trajectory collection featuring vehicle trajectories extracted from 1,140 minutes of drone videos recorded across 12 different locations, capturing a variety of road geometries to facilitate comprehensive analysis and applications. Specifically, we extract 20,000 trajectories in total for both dataset. All the trajectories from both datasets are converted to Frenet state-space [35] using a rule-based motion extractor.

**Baselines.** We compare the performance of ESCIRL against five principled trajectory prediction approaches, including three IRL-based methods that demonstrate better data-efficiency, and two DNN-based methods which excel especially in multi-scenario complex driving conditions. More specifically, the three IRL baselines include Opt-IRL [38], which learns a unified reward function via maximum entropy objective; CIOC [39], which employs a Laplace approximation to estimate rewards in a continuous feature space; and GCL [24], a model-free approach that adopts DNN to learn the reward and policy in an end-to-end manner without predefined heuristics. Additionally, we compare two DNN-based models that have demonstrated superior performance in trajectory prediction tasks: RNN-based models, where we use a CNN-LSTM structure in our implementation (e.g.[40, 41, 42]), and transformer-based (TF-based) models (e.g.[12, 43, 44]). For both model structures, we adopt the default hyperparameter configurations recommended in respective works.

**Metrics.** Existing metrics including averaged feature distance (AFD), average displacement error (ADE), final displacement error (FDE), human probability (HP), and averaged number of collisions (AC) are used in our experiments. Specifically, HP is evaluated by the likelihood that the predicted trajectory aligns with the expert trajectory. It is important to note that not all metrics are applicable to every method. Specifically, AFD is unavailable for GCL, RNN-based, and transformer-based methods, as these approaches do not rely on feature engineering. Similarly, HP cannot be applied to DNN-based methods, as they lack a straightforward evaluation for this metric.

**Results.** To verify ESCIRL's performance under a scarcely labeled dataset which incorporates multiple diverse distributions, we randomly sample 2000 trajectories for both dataset and assess ESCIRL and the five baselines w.r.t. the five aforementioned metrics. The results are shown in Table 1. We can denote that ESCIRL outperforms all the other approaches by a significant margin on both datasets. We attribute the reasons for the strong performance of ESCIRL to the fact that it can efficiently decompose the mixed-scenario expert distributions under limited priors and adaptively

Table 1: Results comparing ESCIRL against three IRL-based and two DNN-based approaches including Opt-IRL, CIOC, GCL, RNN-based, and transformer-based methods. The result denoted as SCIRL refers to the first iteration of ESCIRL without the subsequent evolving process.

| Method | Interaction | | | | | CitySim | | | | |
|---|---|---|---|---|---|---|---|---|---|---|
| | AFD↓ | ADE↓ | FDE↓ | HP↓ | AC↓ | AFD↓ | ADE↓ | FDE↓ | HP↓ | AC↓ |
| Opt-IRL | $1.39_{\pm0.16}$ | $0.76_{\pm0.02}$ | $2.04_{\pm0.13}$ | $8.7\text{e-}3_{\pm0.00}$ | $3.33_{\pm0.33}$ | $2.60_{\pm35.8}$ | $1.86_{\pm0.92}$ | $4.28_{\pm4.19}$ | $1.0\text{e-}2_{\pm0.00}$ | $1.33_{\pm0.33}$ |
| CIOC | $1.68_{\pm0.47}$ | $1.07_{\pm0.05}$ | $3.10_{\pm0.27}$ | $8.9\text{e-}3_{\pm0.00}$ | $2.75_{\pm0.92}$ | $3.32_{\pm6.36}$ | $3.57_{\pm0.79}$ | $7.98_{\pm3.58}$ | $0.01_{\pm0.00}$ | $1.75_{\pm0.92}$ |
| GCL | - | $3.08_{\pm0.01}$ | $7.08_{\pm0.03}$ | $8.5\text{e-}5_{\pm0.00}$ | $5_{\pm2}$ | - | $2.89_{\pm0.02}$ | $5.45_{\pm0.04}$ | $1.3\text{e-}4_{\pm0.00}$ | $1.75_{\pm1.58}$ |
| RNN-based | - | $2.32_{\pm0.15}$ | $5.66_{\pm0.42}$ | - | $4.20_{\pm0.73}$ | - | $2.84_{\pm0.11}$ | $5.98_{\pm0.50}$ | - | $3.33_{\pm0.85}$ |
| TF-based | - | $3.59_{\pm0.54}$ | $7.12_{\pm0.88}$ | - | $7.33_{\pm1.20}$ | - | $4.12_{\pm0.62}$ | $8.30_{\pm0.74}$ | - | $5.88_{\pm1.43}$ |
| *Ours* | | | | | | | | | | |
| SCIRL | $0.46_{\pm0.01}$ | $0.53_{\pm0.00}$ | $1.36_{\pm0.015}$ | $0.02_{\pm0.00}$ | $1.24_{\pm0.50}$ | $1.62_{\pm0.02}$ | $1.09_{\pm0.00}$ | $2.60_{\pm0.01}$ | $0.02_{\pm0.00}$ | $0.45_{\pm0.22}$ |
| **ESCIRL** | $0.45_{\pm0.01}$ | $0.48_{\pm0.01}$ | $1.25_{\pm0.00}$ | $0.01_{\pm0.00}$ | $0_{\pm0.00}$ | $1.51_{\pm0.00}$ | $1.01_{\pm0.00}$ | $2.45_{\pm0.00}$ | $0.02_{\pm0.00}$ | $0.00_{\pm0.}$ |

evolve to optimize the control parameters for a spectrum of reward functions, thus better explaining the diverse expert feature space (low AFD), and consequently resulting in low displacement error (ADE) and averaged collision (AC). Besides, ESCIRL achieves nearly zero safety violation (in terms of AC) on both dataset while other algorithms frequently collide with the environment vehicles. Notably, even without the evolving framework, SCIRL alone still performs better than all the baselines, while ESCIRL further improves the results by iteratively improving the contrastive sets. Additionally, we also note that due to training data scarcity, both the RNN-based and transformer-based method performs poorly on these datasets due to underfitting, which indicates the superiority of ESCIRL in efficiently learning the multi-distribution features under the data scarcity challenge.

## 4 Ablation and Component Interpretation

### 4.1 Generalizability

To better assess the generalizability of ES-CIRL, we conduct an additional experiment where first train a CNN in ESCIRL on 2000 random samples from the INTER-ACTION training set, and then test it on the CitySim test set. The results are reported in Table 2. Results indicate that ES-CIRL outperforms both baselines including the RNN-based and transformer-based methods which have a much larger network to acquire better robustness. We attribute the efficiency and lightweight of ESCIRL to its decoupled training process, where we first obtain the ground truth labels between trajectories and reward function parameters, and fit a CCN to capture this mapping w.r.t. driving context, without directly learning the intricate relationship between the driving context and outcome trajectories.

Table 2: Comparison analysis of generalizability. We train each method on the Interaction dataset and test on the overall CitySim dataset.

| Method | AFD | ADE | FDE | AC |
|---|---|---|---|---|
| Opt-IRL | $2.74_{\pm0.95}$ | $3.56_{\pm0.62}$ | $7.85_{\pm0.92}$ | $4.16_{\pm0.67}$ |
| CIOC | $3.12_{\pm0.86}$ | $3.44_{\pm1.20}$ | $7.36_{\pm1.65}$ | $4.02_{\pm0.97}$ |
| GCL | - | $3.54_{\pm0.75}$ | $7.88_{\pm1.12}$ | $4.55_{\pm1.05}$ |
| TF-based | - | $4.68_{\pm1.42}$ | $8.98_{\pm1.81}$ | $7.11_{\pm1.12}$ |
| RNN-based | - | $3.43_{\pm0.77}$ | $7.23_{\pm1.13}$ | $3.80_{\pm0.70}$ |
| **ESCIRL** | $1.55_{\pm0.07}$ | $1.52_{\pm0.18}$ | $2.85_{\pm0.54}$ | $0.53_{\pm0.75}$ |

### 4.2 Scalability

Besides generalizability, another critical aspect in evaluating trajectory prediction methods is the scalability, which focuses on assessing method's performance when the quantity of available expert demonstrations is limited (data scarcity) as well as very large (data abundance). As shown in Fig. 3, we vary the number of expert demonstrations in training from 500 to 12,000, and observe both in-domain (train and test dataset being the same) and out-of-domain (train

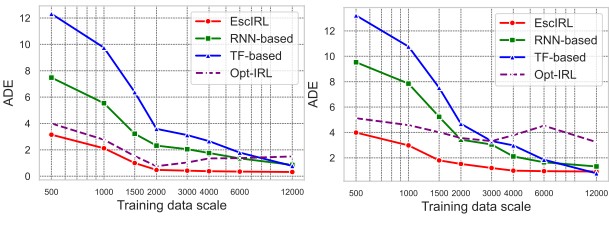

(a) in-domain scalability     (b) out-of-domain scalability

Figure 3: We assess ADE w.r.t. training data scale. We prepare the training set to include random demonstrations from 500 to 12,000 instances from the INTERACTION dataset. In **(a)** these models are tested on the in-domain samples from INTERACTION test set, while in **(b)** they are tested on the out-of-domain Citysim test set.

on INTERACTION and test on CitySim, as in §4.1) performance of the approaches including our proposed ESCIRL. Results demonstrate that ESCIRL has excellent scalability, especially in the data-scarcity scenarios, where ESCIRL significantly outperforms other methods. When the number of available expert demonstrations goes up, ESCIRL is also able to improve its performance. Furthermore, we posit that ESCIRL would further benefit in scenarios of data abundance, particularly when the CCN is enhanced with a large-scale, possibly transformer-based neural network.

### 4.3 Evolving Process

The trajectory prediction performance during the evolving process is shown in Fig. 2b. Both average displacement error (ADE) and average collision (AC) are tracked during this process. The results demonstrate a progressive increase in safe behaviors throughout the evolving process, affirming the robustness of our proposed ESCIRL.

### 4.4 Interpretation Analysis of CCN

Our aim in designing the CCN architecture, as detailed in §2.3, is to maintain the high interpretability characteristic of our proposed ESCIRL. To achieve this, more specifically, we incorporate the attention mechanism [37] to dynamically adjust the focus on various inputs such as map geometry and vehicle kinematic data based on their relevance to the current driving scenario, enabling the ego vehicle to accurately predict and adapt its trajectory.

In this section, we visualize this attention to interpret the model decision making. As shown in Fig. 4, we visualize a scenario from the INTER-ACTION dataset in which the ego vehicle is on a highway, surrounded by multiple vehicles, with the road oriented downwards. We can observe that the CCN utilizes integrated spatial and kinematic maps to discern critical information such as the positions and movements of adjacent and leading vehicles in the same lane, aligning with the vehicle's intent to follow the lane. We believe that this high level of interpretability in the

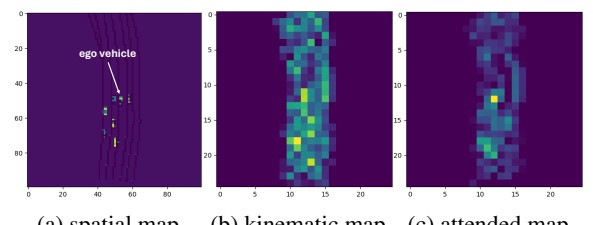

(a) spatial map     (b) kinematic map     (c) attended map

Figure 4: Visualization of hidden layers in CCN. **(a)** and **(b)** depicts the first convolution layer of the spatial and kinematic input. **(c)** depicts the attended mapping by fusing the third hidden layer of both spatial and kinematic streams. Lighter color indicates greater values. The fused map effectively captures the intention of the ego vehicle by attending more on the flanking and leading vehicles in the same lane.

CCN is attributed to its structured design, which may include layered networks that process spatial and temporal data separately to optimize response times and accuracy. Additionally, our unique decoupled training approach allows each component of the system to be trained under varied scenarios, enhancing the model's performance and reliability in complex driving environments.

## 5 Conclusion and Discussion

In this paper, we present ESCIRL, a novel approach that advances trajectory prediction in autonomous driving through evolving self-contrastive IRL. ESCIRL not only enhances prediction accuracy and robustness against diverse driving scenarios, outperforming existing methods by a large margin, but also demonstrates superior generalizability against DNN-based approaches which are known for better generalizability, while maintaining data efficiency as the other IRL-based approaches. In other words, ESCIRL demonstrates advantages from both worlds, which have not been seen in other existing works yet. We attribute the superior performance of ESCIRL to the unique bi-level optimization framework that blends both self-contrastive IRL and an evolving algorithm to offer a compelling balance of the trade-off. Moving forward, future work integrating more complex sensor data would be an interesting way to refine ESCIRL's understanding of more detailed real-world contexts. Additionally, given the successful demonstration of the feasibility for learning generalizable reward functions, it would be meaningful to further explore more unified and adaptive framework for trajectory prediction in the field of autonomous driving.

**Acknowledgments**

We thank Yi Yang at CIDI Lab and all the reviewers for their valuable comments and suggestions.

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

# Appendix

## A  Related Works

### A.1  Driving Behavior Modeling

Driving behavior modeling is essential in autonomous driving, focusing on trajectory prediction and encompassing various methodologies and assumptions [45]. Key areas include behavioral intention prediction (BIP) [46], motion prediction [5, 47], and pattern analysis [48]. BIP predicts future road user actions using historical and contextual data, utilizing parametric models like the intelligent driver model (IDM) [49, 50, 51] and data-driven methods [52, 53, 54]. Parametric models are simple and efficient [55] but may not capture complex dynamics [56], while data-driven approaches handle complex patterns well [46] but struggle with interpretability and adaptability to new data scenarios [57]. Motion prediction enables autonomous vehicles to anticipate other users' movements, enhancing safety and efficiency [47, 58]. It involves physics-based models [59] for simple scenarios and DL methods for complex environments [60, 61]. Pattern analysis extracts features from human driving data to identify driving traits and styles [62], integrating advanced techniques like game theory [63] for deeper insights. Our research advances these concepts by mathematically formulating decision-making in self-driving cars as reward functions, which aid in control and planning, and deriving parameters from driving data to improve trajectory prediction accuracy and interpretability.

### A.2  Trajectory Prediction

Trajectory prediction in autonomous systems is classified into model-based and data-driven approaches [64]. Model-based methods use physics equations and stochastic models [65], such as Bayesian networks [66], Monte Carlo simulations [67], and Gaussian mixture models [68], to predict vehicle movements. While highly interpretable, their application is often limited to less uncertain environments like highways [69]. Data-driven methods, on the other hand, learn from historical and spatial data using recurrent [70, 71], convolutional [40], graph [72, 73] neural networks, transformers [11, 12, 13], and generative adversarial networks [74], which excel in handling complex temporal and spatial data. Recent studies also explore self-supervised contrastive learning [34], proven effective in supervised [75] and unsupervised [34, 76, 77] settings, to improve trajectory prediction. This work introduces an action-based contrastive loss that incorporates pedestrian action data, enhancing the accuracy and context-sensitivity in complex environments.

### A.3  Robust IRL

Both model-based and data-driven trajectory prediction methods struggle with complexity and require extensive data, often producing only average predictions that miss the variability in human driving [78]. Multimodal methods address this by considering multiple possible paths [79, 80, 81], but they can be unstable and overlook specific driving characteristics. IRL and imitation learning aim to more closely mimic human driving by learning from observed behaviors, yet they are limited by their reliance on environmental models for simulating interactions [22, 82, 33]. Robust IRL, developing from robust Markov Decision Process (MDP) frameworks [83, 84, 85, 86, 87], focuses on enhancing decision-making robustness. Our work diverges from traditional approaches by addressing dynamics mismatches rather than just learner-expert mismatches, a significant issue in Generative Adversarial Imitation Learning (GAIL) [88] and its variants [89, 90]. On the other hand, ESCIRL offers a robust, generalizable solution that enhances policy robustness and applicability by handling these mismatches effectively.

## B  Experiment Settings

### B.1  Maximum Entropy Inverse Reinforcement Learning (MaxEnt-IRL)

Recalling the definition in Eq. (1), $\sigma$ and $u$ represent the states and actions of the vehicle respectively, we formulate the dynamics $f(\cdot)$ of each vehicle as defined in Eq. (1). A driving trajectory $\xi$ within the spatial-temporal domain comprises a sequence of states and actions, such that $\xi = \{\sigma^0, u^0, \sigma^1, u^1, \ldots, \sigma^{N-1}, u^{N-1}\}$, where $N$ denotes the length of the planning horizon. Given a set of driving demonstrations $\mathcal{D}_p = \{\xi_i\}$, indexed by $i = 1, 2, \ldots, N$, the maximum-entropy

IRL [29] problem seeks to infer the underlying reward function that maximizes the likelihood of these expert demonstrations, under the assumption that trajectories are exponentially more likely with higher cumulative rewards. This assumption is formalized by the Boltzmann noisily-rational model $P(\xi)$ [31] as shown in (10), where $\omega$ is the parameters of reward function $R$ and $\beta$ is a hyper-parameter to specify demonstration proximity [30], which we consistently adopt $\beta = 1$ in this study. Notably, while previous works typically partition datasets into particular scenarios such as in-lane following [22], lane-changing [32], and merging [33] to assume a stable reward function across demonstrations, we engage with the complete dataset instead. In our study, the dataset contains multi-modal driving policies with distinguished distributions, and we aim to learn a continuous controller network to handle diverse driving scenarios presented in the dataset.

Our methodology employs a linear-structured reward function that is characterized by a specifically chosen feature space $\mathcal{F}(\cdot)$, which is defined in relation to the trajectories $\xi$:

$$R(\xi; \omega) = \omega^T \mathcal{F}(\xi) \tag{9}$$

As a result, the probability, or likelihood, of the demonstration set is given by:

$$P(\mathcal{D}_p|\omega) = \prod_{i=1}^{N} \frac{e^{\beta R_\omega(\xi_i)}}{Z_\omega} = \prod_{i=1}^{N} \frac{1}{Z_\omega} e^{\beta R_\omega(\xi_i)} \tag{10}$$

In this context, $Z_\omega$ denotes the set of all feasible trajectories that correspond to the initial and terminal conditions delineated by $\xi$. The primary objective is to ascertain the optimal parameter vector $\omega^*$ that enhances the average log-likelihood of the observed demonstrations, as shown below:

$$\omega^* = \arg \max_\omega \frac{1}{N} \log P(\mathcal{D}_p|\omega) \tag{11}$$

$$= \arg \max_\omega \frac{1}{N} \sum_{i=1}^{N} \log P(\xi_i|\omega) \tag{12}$$

From (4) and (5), we can see that the key step in solving the optimization problem in (5) is the calculation of the partition factors $Z_\omega$. In sampling-based methods, $Z_\omega$ for each demonstration is approximated via the sum over samples in the sample set $\{\xi_i^m\}$, $m = 1, 2, \ldots, M$, and $\xi_i$ is denoted as $\xi_i^0$:

$$Z_\omega \approx \sum_{m=0}^{M} e^{\beta R_\omega(\xi_j^m)} \tag{13}$$

Thus, the objective function in (5) becomes:

$$\begin{aligned} L(\omega) &= -\frac{1}{N} \sum_{i=1}^{N} \log P(\xi_i|\omega) \\ &= -\frac{1}{N} \sum_{i=1}^{N} \log \frac{e^{\beta R_\omega(\xi_i)}}{\sum_{m=0}^{M} e^{\beta R_\omega(\xi_i^m)}} \\ &= -\frac{1}{N} \sum_{i=1}^{N} \left( \beta R_\omega(\xi_i) - \log \sum_{m=0}^{M} e^{\beta R_\omega(\xi_i^m)} \right) \end{aligned} \tag{14}$$

The derivative is thus given by:

$$\nabla_\omega L = \frac{\beta}{N} \sum_{i=1}^{N} \left( \mathcal{F}(\xi_i) - \hat{\mathcal{F}}(\xi_i) \right) \tag{15}$$

$$\hat{\mathcal{F}}(\xi_i) = \frac{\sum_{m=0}^{M} e^{\beta R_\omega(r_i^m)} \mathcal{F}(\xi_i^m)}{\sum_{m=0}^{M} e^{\beta R_\omega(\xi_i^m)}} \tag{16}$$

where $\mathcal{F}(\xi)$ defines the expected feature counts over all samples given $\omega$.

Note that an additional $l_1$ regularization over the parameter vector $\omega$ is introduced in the training process to compensate for possible errors induced via the selected set of features.

## C Feature Design for EscIRL

For any trajectory $\xi = \sigma_i{}_{i=0}^{T-1}$ with $T$ frames of states, where $\sigma_i = [x_i, y_i, \theta_i, s_i, d_i, v_i]$, we project it into feature space as follows.

### C.1 Features for comfort

$$f_1 = \frac{1}{T-1} \sum_{i=0}^{T-2} (v_{i+1} - v_i)^2 \tag{17}$$

$$f_2 = \frac{1}{T-2} \sum_{i=0}^{T-3} (v_{i+2} - v_i)^2 \tag{18}$$

We introduce $f_1$, $f_2$ to prevent dramatic velocity changes. Essentially, $f_1$ calculates the average sum of squared differences in velocity between consecutive states, which is a measure of the smoothness of velocity over time. $f_2$ is similar to $f_1$, but it takes the average squared difference in velocity over every two time steps instead of every one. This means it measures the change in velocity over a longer time span. By minimizing these two features, we can avoid jerky movements which may cause uncomfortable riding experience.

$$f_3 = \frac{1}{T-1} \sum_{i=0}^{T-2} (\mathrm{Wrap}(\theta_{i+1} - \theta_i))^2 \tag{19}$$

$$\mathrm{Wrap}(\theta) = \theta - 2\pi \lfloor \frac{\theta + \pi}{2\pi} \rfloor \tag{20}$$

$f_3$ sums up the average squared change of heading angles over consecutive steps. Here, the Wrap function ensures the heading angle change is bounded in $[-\pi, \pi]$. It is minimized to ensure a smooth trajectory.

### C.2 Features for efficiency

$$f_0 = \frac{1}{T} \sum_{i=0}^{T-1} v_i{}^2 \tag{21}$$

$f_0$ sums up the squared velocity over the whole trajectory. By minimizing $f_0$, the model penalizes abnormally high velocity, thus ensuring a more fuel efficient and environment friendly driving dynamic.

$$f_6 = \frac{1}{T} \sum_{i=0}^{T-1} (\frac{v_i - v_{\mathrm{limit}}}{v_{\mathrm{limit}}})^2 \tag{22}$$

$v_{\mathrm{limit}}$ is the velocity limitation of the current road. To ensure the high efficiency of trajectory planning, we may want the vehicle's speed to be fast without violating traffic rules. By minimizing $f_6$, we can ensure the vehicle's speed is close to the speed limit.

$$f_4 = \frac{1}{T-1} \sum_{i=0}^{T-2} (s_{T-1} - s_i)^2 \tag{23}$$

$$f_5 = \frac{1}{T-1} \sum_{i=0}^{T-2} (\mathrm{Wrap}(\arctan(\frac{y_{T-1} - y_i}{x_{T-1} - x_i}) - \theta_i))^2 \tag{24}$$

$$f_9 = \frac{1}{T} \sum_{i=0}^{T-1} I_i^{\mathrm{L}} \tag{25}$$

$$f_{10} = \frac{1}{T} \sum_{i=0}^{T-1} I_i^{\text{R}} \tag{26}$$

$I^{\text{L}}$ and $I^{\text{R}}$ represent the ego vehicle whether to merge left or right, according to its destination.

$$f_{11} = \frac{1}{T} \sum_{i=0}^{T-1} (e^{d_i^{\text{targetL}}} - 1)(\max(s^{\text{margin}} - s_i^{\text{end}}, 0) + 1) \tag{27}$$

$$f_{12} = \frac{1}{T} \sum_{i=0}^{T-1} (e^{d_i^{\text{targetR}}} - 1)(\max(s^{\text{margin}} - s_i^{\text{end}}, 0) + 1) \tag{28}$$

$d^{\text{targetL}}$ and $d^{\text{targetR}}$ are the distance from the ego vehicle to the left or right road's center line if it should merge left or right. $s^{\text{margin}}$ is a constant distance to measure the urgency of changing lanes. $s^{\text{end}}$ is the distance from the current pose to the endpoint of the current road.

### C.3 Features for safety

$$f_7 = \frac{1}{T} \sum_{i=0}^{T-1} (d_i)^2 \tag{29}$$

$$f_8 = \frac{1}{T} \sum_{i=0}^{T-1} (\text{Wrap}(\theta_i - \theta_{\text{road}}))^2 \tag{30}$$

$f_8$ measures the deviance of vehicle's heading angle from the road's angle. Minimizing this feature keeps the vehicle on the right track.

$$f_{13} = \frac{1}{T} \sum_{i=0}^{T-1} e^{-|d_i^{(E-S_F)}|} \tag{31}$$

$$f_{14} = \frac{1}{T} \sum_{i=0}^{T-1} e^{-|d_i^{(E-S_L)}|} \tag{32}$$

$$f_{15} = \frac{1}{T} \sum_{i=0}^{T-1} e^{-|d_i^{(E-S_R)}|} \tag{33}$$

$d^{(E-S_F)}$, $d^{(E-S_L)}$, and $(E - S_R)$, is the distance from the ego vehicle to the nearest surrounding vehicle in front, left, or right region respectively, and denoted as $\inf$ if no surrounding vehicle exits in such region.

$$f_{16} = \frac{1}{T} \sum_{i=0}^{T-1} e^{-|d_i^{(E-C_F)} - d_i^{(S_F-C_F)}|} \tag{34}$$

$$f_{17} = \frac{1}{T} \sum_{i=0}^{T-1} e^{-|d_i^{(E-C_L)} - d_i^{(S_L-C_L)}|} \tag{35}$$

$$f_{18} = \frac{1}{T} \sum_{i=0}^{T-1} e^{-|d_i^{(E-C_R)} - d_i^{(S_L-C_R)}|} \tag{36}$$

A collision point between the ego vehicle and the nearest surrounding vehicle in the front, left, or right region would be predicted respectively. We chose the Constant Velocity model as our prediction

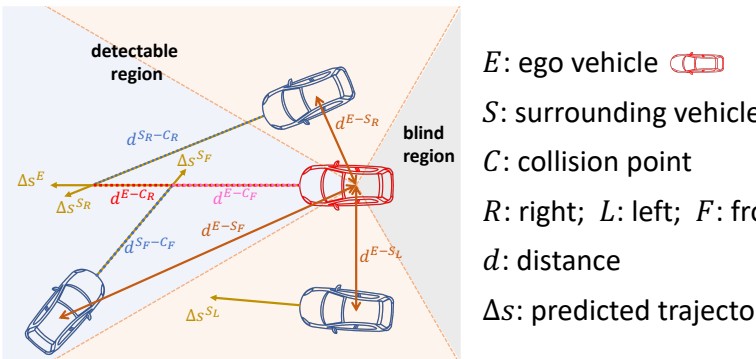

Figure 5: Interactive feature design

.

method, which is the simplest method. $d^{(E-C_F)}$, $d^{(E-C_L)}$, and $d^{(E-C_R)}$ is the distance from ego vehicle to the collision point with the nearest surrounding vehicle in the front, left, and right region, while $d^{(S_L-C_F)}$, $d^{(S_L-C_L)}$, and $d^{(S_L-C_R)}$ is the distance of the surrounding vehicle. A sample is shown in Figure 5.

$$f_{19} = \frac{1}{T} \sum_{i=0}^{T-1} N_i^{\text{coll}} \tag{37}$$

$N^{\text{coll}}$ is the number of collisions between ego vehicle and surrounding vehicles.

## D Continuous Controller Network (CCN)

### D.1 Structure Design

Given a trajectory $(\tau_i)$, a parameter $\omega_i$ would be calculated through Contrastive IRL. We designed a Continuous Control Network (CCN) to map the relationship between the first states of trajectories and CIRL parameters, which provides a continuous mapping method for any given state.

Let $\hat{\mathcal{D}} = \{(\sigma_i^0, \omega_i)\}_{i=1}^N$ be a dataset for state-parameter pairs, where the $\sigma_i^0$ is the initial state of each trajectory. We first encode the initial state into a multi-channel context map $\mathcal{M} \in \mathcal{R}^{c \times 200 \times 200}$ with $c$ channels and size $(200, 200)$. The first channel is road information near the ego vehicle; the second channel is the spatial information about the ego vehicle and surrounding vehicles, containing their positions, headings, and shapes; other channels are the kinematic information about the ego vehicle and surrounding vehicles, containing velocity, acceleration, distance to road center, etc. The kinematic information about the ego vehicle is also collected as ego kinematic data.

The structure of the Continuous Control Network is shown in Figure 2a. The kinematic channels of the context map are fed into two layers' 2D convolutional neural network with kernel size $(6, 6)$, stride 2, and padding 2, and a 2D max pooling layer in the middle with kernel size $(2, 2)$ and stride 2. The output of the last layer is 32 channels with the size $(25, 25)$, denoted as an attention map. The context map's road channel and spatial channel are fed into three layers' 2D convolutional neural network with kernel size $(4, 4)$, stride 2, and padding 1. The output of the last layer (original map) is also 32 channels with the size $(25, 25)$ and fused with the attention map by Hadamard Product. The original and fusion maps are extracted into feature vectors through the fully connected layers. The ego kinematic data is also extracted into a feature vector and concatenated with the other two feature vectors. Finally, the prediction of reward parameters $\omega^{pred}$ mapping to the given initial state is predicted from the final FC layer by feeding the concatenated feature vector as input.

To update the parameters of CCN, the loss function is denoted as follows:

$$\mathcal{L}_{\text{CCN}}(\omega^{pred}|\omega) = \frac{1}{N} \sum_{i=1}^N \log(\cosh(|\omega_i^{pred} - \omega_i|)) \tag{38}$$

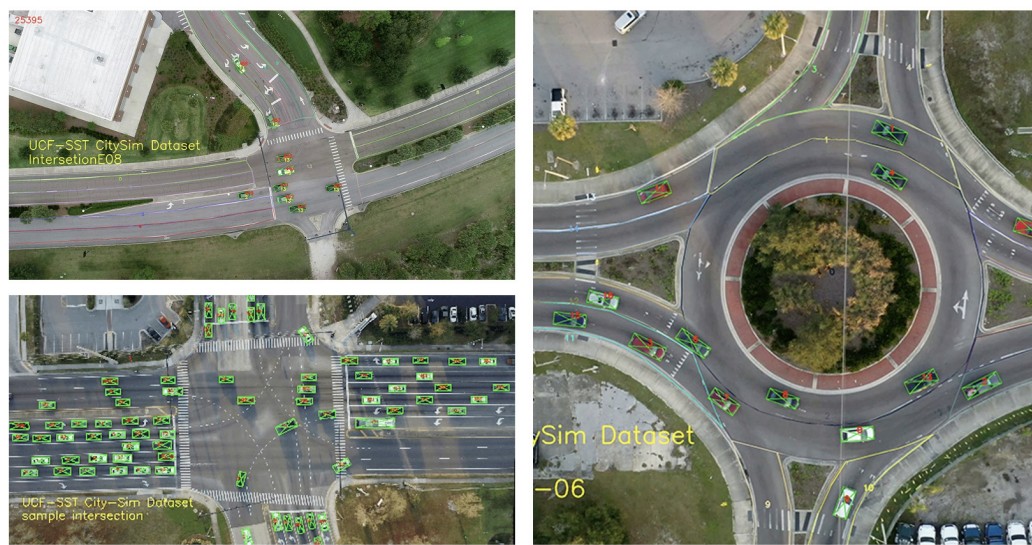

Figure 6: The interactive scenarios of CitySim dataset (e.g. intersection, roundabout) that we investigate in this study [26].

## D.2 Dataset

While a vast array of datasets exists for autonomous vehicles, they often encompass a restricted range of real-world scenarios. To address the limitations in variety and depth of data, particularly acute in regions with less technological infrastructure, our model incorporates a contrastive process that enhances its capability to handle diverse real-world driving conditions with limited feature sets.

To realize this, the datasets selected for training are characterized by extensive variability and a rich set of annotations necessary for accurately generating labels for subsequent training stages. Adhering to these selection criteria, we have curated a suite of datasets that provide comprehensive coverage of the myriad situations encountered on the roads.

**INTERACTION.** The INTERACTION dataset encompasses four categories of interactive driving scenarios collected from 11 different locations, provides a rich, diversified backdrop that reflects the variability of driving behaviors across the globe with driving scenarios from different countries and cultural contexts. This inclusivity ensures that models developed or tested against this dataset can be more easily adapted to the unique conditions of developing countries, mitigating the bias often found in datasets predominantly sourced from high-resource environments.

**CitySim.** The selection of the CitySim dataset is predicated on its unparalleled contribution to safety-oriented research and applications within the domain of autonomous driving, particularly under the constraints of developing countries where safety-critical events and infrastructure limitations pose unique challenges. CitySim's comprehensive and meticulously curated vehicle trajectory data set the stage for innovative machine learning approaches that are both resource-efficient and highly applicable to real-world scenarios.

These datasets are instrumental in overcoming the prevalent data scarcity and contribute to a more robust and versatile model training process. They ensure that the resulting algorithms are capable of interpreting and reacting to a wide array of traffic scenarios, making the advancements in autonomous vehicle technology more accessible and effective across different regions.

