# OpenReview forum: "EscIRL: Evolving Self-Contrastive IRL for Trajectory Prediction in Autonomous Driving"
_robot-learning.org/CoRL/2024/Conference — CoRL 2024_

### Official Review · Reviewer_iJ2t · 2024-07-20
**Novel idea with compelling results in data-scarce trajectory prediction**

**Originality:** 3
**Technical Quality:** 3
**Clarity Of Presentation:** 3
**Potential Impact:** 3
**Recommendation:** 3
**Confidence:** 3

**Review:**

Strengths:
- The paper is generally well written and generally presents ideas clearly.
- The algorithm is well motivated, and presents a novel approach to apply IRL to settings where the assumption that the demonstrations are explained by a single reward function does not hold.
- The experiments are comprehensive and successfully address the motivation of the paper, demonstrating the performance in data-scarce regimes, the improved generalization to held-out datasets, and how the algorithm scales to larger amounts of data. The results are compelling, showing impressive generalization in low-data settings while still scaling well with larger data. I would have liked to see this scaling experiment extended beyond the 12000 sample limit, as it appears that the transformer based HOME was still showing strong returns on scaling the dataset.

Weaknesses:
- The evolving framework was not explained very clearly. I would have liked to see the strategy laid out clearly in the algorithm block, but it was not clear to me that Algorithm 1 included it. What exactly is being mutated and crossed over? How is the number of contrastive sets chosen?
- The paper is presented as targeting trajectory prediction, yet Algorithm 1 describes learning a policy $\pi$, which is not discussed in the body of the paper.
- Driving trajectories may be multimodal, i.e., in a single state, there may be equally valid choices for future behavior of vehicles. For this reason, many trajectory forecasting papers sample multiple trajectories from the model and present minADE/FDE metrics over the top $k$ samples, to demonstrate how well the forecasts cover the range of possible behavior. Adding these metrics would strengthen the paper, as would a discussion of how this approach handles multimodality in future trajectory prediction.


Comments:
- Define $\mathcal{F}$ used in equation (2).
- Section 2.3, what does it mean for $\xi_{i}^- = \emptyset$? If it means that the negative sample set is chosen to be the empty set, then consider using notation to define these sets explicitly (rather than just defining the elements $\xi_i^-$ and cardinality $K^-$.

**Quality Of The Limitations Section:**

1

**Questions For Rebuttal:**

- Could you describe in more detail how the evolving strategy works?
- How sensitive is the approach to the choice of $K^+$ and $K^-$? How are these values chosen?

**Robotics Focus:**

3

**Summary Of Paper:**

This paper presents EscIRL, an algorithm for trajectory prediction which aims to combine the data-efficiency of IRL with the robustness and generalizability of supervised learning approaches to trajectory prediction. The approach models a dataset of trajectory demonstration as a set of distinct policies, and aims to jointly learn the specific sets, as well as a reward function describing behavior in each set. In contrast to previous work, the authors make no assumptions on how to define these sets, but rather learn them directly on data using an evolutionary framework.

**Summary Of Recommendation:**

I found the idea novel, results compelling, and would support acceptance assume authors are able to clarify details of their evolutionary approach.

---

### Official Review · Reviewer_vT2J · 2024-07-31
**ESCIRL: Evolving Self-Contrastive IRL for Trajectory Prediction in Autonomous Driving**

**Originality:** 3
**Technical Quality:** 4
**Clarity Of Presentation:** 4
**Potential Impact:** 3
**Recommendation:** 3
**Confidence:** 3

**Review:**

The paper introduces the Evolving Self-Contrastive Inverse Reinforcement Learning (ESCIRL) framework to address data scarcity and generalizability issues in the state-of-the-art DNN and IRL methods in trajectory prediction tasks. ESCIRL is a decoupled bi-level training framework with a self-contrastive IRL (SCIRL) module at the inner level and an evolving loop at the outer level. Experimental results on the CitySim and INTERACTION datasets demonstrate that ESCIRL outperforms state-of-the-art DNN and IRL methods by an average of 68.2%.

### Quality:
The paper explained the limitations of the existing DNN- and IRL-based approaches and proposed a well-thought-out framework to address these common challenges. The author provided details of the workflow, the algorithm, and the mathematical description for each component. The proposed solution was compared against five SOTA trajectory prediction approaches on two multi-scenario datasets and evaluated with multiple metrics. The results demonstrated the effectiveness of the proposed solution. In addition, the author conducted a detailed ablation study and interpreted each component. The findings are convincing.

(weakness) It would be helpful to highlight the incremental technical differences between the proposed and existing solutions in more detail, especially on what the SOTA solutions have already achieved.

### Clarity:
the paper is well-written and easy to follow.

### Originality:
Although the components in the framework have been studied in various applications, according to the authors, the proposed setting has not been seen in the state-of-the-art solutions in trajectory prediction tasks.

(weakness) It would be helpful to provide more literature on the proposed framework and components and elaborate on how previous works inspired the idea.

### Significance:
Based on the difference between the proposed SOTA solutions and the experiment results, the proposed solution seems to improve trajectory prediction tasks significantly.

(weakness) The authors need to justify why such a training framework combining SCIRL and evolving loop may bring benefits to other IRL applications. Has a similar framework been seen in other applications?

**Quality Of The Limitations Section:**

2

**Questions For Rebuttal:**

1. What are the limitations of the proposed solution, e.g., complexity, training process?
2. Has a similar framework been seen in other Robotics or ML applications? Are there any literature on the framework?
3. Further highlight and elaborate on the novel parts of the proposed framework (Fig 1.) vs. existing solutions.

**Robotics Focus:**

3

**Summary Of Paper:**

The paper introduces Evolving Self-Contrastive Inverse Reinforcement Learning (ESCIRL), a novel approach that integrates DNN and IRL for trajectory prediction in autonomous driving. ESCIRL employs a decoupled bi-level training framework, with a self-contrastive IRL module at the inner level and an evolving loop at the outer level. The approach aims to address data scarcity and generalizability issues inherent in DNN and IRL methods. Experimental results on the CitySim and INTERACTION datasets demonstrate that ESCIRL outperforms state-of-the-art DNN and IRL methods by 82.5% and 58.7% on average.

**Summary Of Recommendation:**

The proposed innovative framework presents a significant advancement in trajectory prediction for autonomous driving by addressing the limitations of current DNN and IRL methods, particularly in data efficiency, robustness, and generalizability. The experiment setting, results, ablation study, and interpretation are convincing.

---

### Official Review · Reviewer_RjKv · 2024-08-03
**This paper is generally well-written and structured. However, there are several areas that need attention and clarification.**

**Originality:** 4
**Technical Quality:** 3
**Clarity Of Presentation:** 4
**Potential Impact:** 3
**Recommendation:** 3
**Confidence:** 4

**Review:**

This paper is generally well-written and structured. However, there are several areas that need attention and clarification.

There is a formatting error in lines 77-79 when referencing sections 2.1-2.3 (§2.2). Please correct this to ensure proper referencing. Additionally, in line 82, why the function is defined as  𝑓(⋅)

In line 119, provide an explanation for why and how the parameter γ is set to 1. It is important to justify this choice to help readers understand its significance and impact on the results.

Furthermore, in line 124, clarify why "higher probability in the positive set than the negative set" is chosen. Explain the rationale behind this decision and its implications for the overall methodology.

To improve the flow of the paper and aid in the understanding of the content discussed, include Figure 2(a) before Section 2.3.

Additionally, in lines 190-201, provide a detailed explanation of how the baseline algorithms are chosen for comparison. Discuss the criteria used for their selection and why these particular algorithms are relevant to the study.
 Finally, offer a thorough explanation and references for the metrics chosen to evaluate the algorithms in lines 202-207.

**Quality Of The Limitations Section:**

1

**Questions For Rebuttal:**

Please provide an explanation of how the baseline algorithms are chosen for comparison. Are there any other algorithms that could perform better?

Comments by the authors are acceptable. But suggested changes need to be fully addressed if accepted for the publication,

**Robotics Focus:**

3

**Summary Of Paper:**

This papert introduce a novel decoupled bi-level training framework that iteratively learns robust reward functions from complex multi-scenario expert demonstrations. This innovative approach has been experimentally verified using a public dataset

**Summary Of Recommendation:**

This paper is well-written and introduces a new algorithm that has performed well compared to previous work. However, the limitations are not well discussed. Although there was a discussion about generalizability, the algorithm needs to be assessed on more datasets to make a stronger argument about the generalizability of the proposed algorithm

---

### Decision · Program_Chairs · 2024-09-04

**Decision:**

Accept

**Comment:**

The strengths of this paper are clarity of overall presentation, motivation, and presentation of results.  There were several weaknesses identified by the reviewers in the submission. However, during the rebuttal stage the authors have successfully addressed the reviewers' concerns and clarified the details improving the quality of the paper.